# Readability Metrics for Machine Translation in Dutch: Google vs. Azure & IBM

**Chaïm van Toledo[1,*]**, **Marijn Schraagen [1]**, **Friso van Dijk [1]**, **Matthieu Brinkhuis [1]** and **Marco Spruit [2,3]**

1. Department of Information and Computing Sciences, Utrecht University, Princetonplein 5, 3584 CC Utrecht, The Netherlands
2. Department of Public Health and Primary Care, Leiden University Medical Center (LUMC), Albinusdreef 2, 2333 ZA Leiden, The Netherlands
3. Leiden Institute of Advanced Computer Science, Leiden University, Niels Bohrweg 1, 2333 CA Leiden, The Netherlands
* Correspondence: c.j.vantoledo@uu.nl

**Abstract:** This paper introduces a novel method to predict when a Google translation is better than other machine translations (MT) in Dutch. Instead of considering fidelity, this approach considers fluency and readability indicators for when Google ranked best. This research explores an alternative approach in the field of quality estimation. The paper contributes by publishing a dataset with sentences from English to Dutch, with human-made classifications on a best-worst scale. Logistic regression shows a correlation between T-Scan output, such as readability measurements like lemma frequencies, and when Google translation was better than Azure and IBM. The last part of the results section shows the prediction possibilities. First by logistic regression and second by a generated automated machine learning model. Respectively, they have an accuracy of 0.59 and 0.61.

**Keywords:** quality estimation; SQUAD 2.0; machine translation; English to Dutch quality estimation

## 1. Introduction

Translating from a source language to a target language is a difficult task. An author must be competent in both the source and the target language [1]. An excellent assistant for this task is machine translation (MT). MT is faster than any human. The speed of MT is a huge advantage, but what good is the speed if you can't estimate the quality? With a few words, manually estimating the quality is easy. However, weighting the translation quality manually is quite difficult when there are a bulk of documents.

Two automatic options for estimating MT quality are (1) machine translation evaluation (MTE) and (2) quality estimation (QE) [2]. With the option of MTE, the methods demand a human-translated text to measure how close the MT is to a human translation. With metrics like BLEU (bilingual evaluation understudy), a score shows how close the MT comes to human translation. Nevertheless, for every new translation, new human translation tasks are needed to measure to which extent the MT comes to a human translation.

With QE, the need for reference texts is gone. Although, when QE came from a machine learning perspective, it is needed for training purposes. The outcome of QE can be binary: good or bad. Additionally, also an estimation of how good the translation is. In several attempts for QE, data are necessary to build a QE model. Much data come from the Conference on Machine Translation (WMT) [3], which has multiple datasets. Other domains provide, for example, legal QE datasets [4].

In this research, the focus is on readability and text metrics. Text metrics score text on different axes. With those metrics, readability can be scored. The domain of readability has multiple measurement methods. For the Dutch language, there is a tool to measure many facets of text, namely T-Scan [5]. Simple readability metrics like sentence lengths and more complex measurements such as word probability scores can be calculated.

When reading a text, a reader should experience coherence between words. In a coherent text, readability and fidelity should go together. Fidelity indicates how accurate semantically the translation is [6]. The task and purpose of this paper are quite simple: predict if the Google translation is better than the translations of Azure and IBM using readability metrics. Another purpose is to find text analytic features corresponding to the previously written task.

Hence the research question: Is it possible to predict if Google is better than Azure and IBM with T-Scan readability features? Several sub-questions divide the research question: Which T-Scan features can help score the best machine translation? Which combinations of features of T-Scan will perform the best prediction?

We present the potential of readability features in combination with QE as an alternative method to estimate QE in Dutch. In the experimental setting, 213 English sentences are translated to Dutch with Microsoft Azure's Translator API, IBM Language Translator and Google Translator v3. We provide these translated sentences as a new dataset for further research: These sentences are humanly ranked and are further analysed by T-Scan. Logistic regression analysis examined the correlation between the Google translations and the T-Scan features. The prediction possibilities of such a model are further explored with a logistic regression model and a Gradient Boosting Classifier, which was generated by automated machine learning (AutoML).

The paper is structured as follows: the related section will elaborate on the readability, the text analysing tool T-Scan and quality estimation. In the methods section, the setup of the experiment is explained. The results show the logistic regression summary matrix and the classification matrix. The last part of the work is the discussion, future work, and conclusion. All code and data are made publicly available (Code and data on https://github.com/7083170/Readability-metrics-for-machine-translation (accessed on 20 March 2023)).

## 2. Related Work

In this section, we first explain readability, second readability tooling, third examining T-Scan, fourth machine translation and fifth and last, QE.

### 2.1. Readability

Readability is an essential subject in text analysis. A clear definition of readability is the ease a reader can understand a text [7]. The definition can be extended by adding the ease of understanding the author's writing style. Important to address is that fonts and layout are not parts of readability [8].

For analysing readability, many features are available. These features assume a particular knowledge of the reader [9]. Humans learn words from other humans [10]. Humans learn by talking with other people and by reading texts. Many readability features use this assumption, namely the degree of word knowledge and the degree of word predictability [11].

### 2.2. Readability Tooling

There are a tremendous number of text analysing tools. Most of these text analysing tools say something about readability. For example, the number of words in a sentence: longer sentences are harder to read. There are many of these features, and some of the tools are also a combination of features. Instead, this part focuses on some of the critical features.

As readability stands for ease of understanding, various features can be measured by analysing the texts. For example, word prevalence, entropy, perplexity, word count, particular character count, word probability, morpheme count, word frequencies, named entity recognition, sentence analysing features and many more. Some tooling uses more than 200 features [12], or even more than 400 formulas [5].

Which of those features are relevant? With word frequencies, the assumption is that a reader has encountered words before. In Dutch, there are, for example, the Staphorsius frequency [13] and the newer Basilex corpus [14]. These corpora hold a large part of words

from schoolbooks. The proliferation of these words is due to compulsory education in the Netherlands. Hence, the chance that readers saw these words earlier in their lives is high. Therefore, these frequencies can indicate some part of readability [11].

Another attempt for measuring readability is with word prevalence. The feature represents the number of people who know the word [15]. These kinds of corpora are created by asking people whether they know the word.

The uncertainty of language is measured with entropy. "Word entropy is a 'forward-looking' metric and models the degree of the listener's or reader's uncertainty about the upcoming word given the words encountered so far" [16] (p. 285). The entropy rises when a sentence contains many words that do not occur in a common language model. On the other hand, perplexity measures the possibilities of a language model. "Perplexity is defined as the exponential transformation of surprisal" [16] (p. 285).

### 2.3. T-Scan

T-Scan is a tool which provides 457 different kinds of measurement features for a Dutch document [17]. The 457 features are divided into nine groups. The first group is the default features, like the number of sentences and paragraphs. The second group is word difficulty and contains 88 features, like word frequencies, number of morphemes, characters per word, and word prevalence. The third group is sentence complexity and holds 73 features, about words per sentence, subordinate clauses and more.

The fourth group is reference coherency and lexical diversity. The group contains 26 features. These features are context words per subordinate clause or the number or the density of lemmas compared to a previous sentence. The fifth group, relational coherency and situation model metrics, stores 40 features. These features vary in the density of time and emotional and causal words.

In the sixth group, the semantic classes, concreteness and generality of texts are measured. The group covers a high number of 133 features. The features vary from the proportion of the nouns with different kinds of references to the density of general adverbs. The seventh group contains five features about personal elements. The eighth group is about other lexical information and includes 76 features about entity names like products, events, persons, and organisations, based on the Dutch named entity recognition and word analysing tool "Frog" [18].

The last group are the probability metrics, with 16 features. With measurements such as perplexity, entropy, and the logarithm of the trigram probability.

T-Scan is provided with other Dutch text analytics tools at the Lamachine web services. The tool is used for testing text comprehension [19], lexical en sentence complexity [20], analysing text genres [21] and other textual-related research for the Dutch language. T-Scan output divides document, paragraphs, and sentence-oriented scores in a CSV.

### 2.4. Machine Translation

Machine translation moved from a statistical machine translation (SMT) approach to a neural machine translation (NMT) approach [22]. SMT uses machine learning models to translate source texts to a target language [23]. SMT can build new sentences with a (large) parallel text corpus. NMT replaced most of the SMT approaches in cases when there is a large corpus.

An NMT approach uses two recurrent neural networks (RNN) to consume the source text and the second to create a target text. NMT gives better translation quality than SMT, but when in a low-resource situation, SMT outperforms the NMT approach [22]. Not finding rare words or not translating all words are sometimes failures in the early attempts of NMT [24].

Other attempts in comparing NMT approaches were made in [25]. In this part between Azure's Translator, Deepl and Google Translation. In this case, the parliamentary texts from French to English were compared. The outcome was that Azure and Deepl didn't have significant differences; on the other hand, Google Translation had fewer collocational bigrams.

Google presents its translation machine as Google's Neural Machine Translation system. The system uses long short-term memory RNNs [24]. Further, it breaks words into wordpieces (later also used in BERT models [26]).

Microsoft Azure's Translator API, IBM Language Translator and Google Translator v3 all use the NMT approach [27–29].

### 2.5. Quality Estimation

The goal of QE for MT is to evaluate the quality of the translated text without reference texts [30]. The estimation can be binary (good or bad) or with a discrete or continuous score. QE applications can be different: identifying bad translations, selecting the best translation between multiple MT providers, and indicating time for post-editing an MT. QE is not necessarily worse than MTE. Results from attempts show for specific languages (Spanish-English, English Spanish, no significantly different results (QE score) than with reference texts (Meteor score) [30].

For example, the approaches for QE are executed with support vector machines and with neural network approaches [31]. A large amount of annotated data is needed to give insights for QE. However, there are also unsupervised methods [32]. Some of these ideas are worked out in the open-source community. The open-source framework of OpenKiwi provides a machine-learning approach for QE [33]. The approach requires training material for estimation. OpenKiwi is built in python and uses PyTorch. Another approach is QuEst [34]. Like OpenKiwi, QuEst is also a python-built framework. QuEst makes use of the WMT data for QE.

QE now uses data provided by WMT but can also use revision tracking, readability, comments and publication acceptance metrics [35,36]. One of the state-of-the-art QE frameworks (according to the WMT QE shared task) is TransQuest [37]. Besides the WMT datasets, there is also a framework and datasets for English-to-Dutch, -French and -Portuguese in the legal domain [4]. However, the English-to-Dutch dataset is built on legal documents instead of general wiki texts. With the lack of *-to-Dutch datasets, this paper tries to combine readability metrics with QE.

## 3. Methods

A research path is taken to estimate when Google translation is better than Azure and IBM. The Figure 1 below sketches the steps. These steps can be categorised into two paths, namely (1) the construction of the dataset and (2) the analysis part, where the automatic text analysis from T-scan is compared with the best-worst scale.

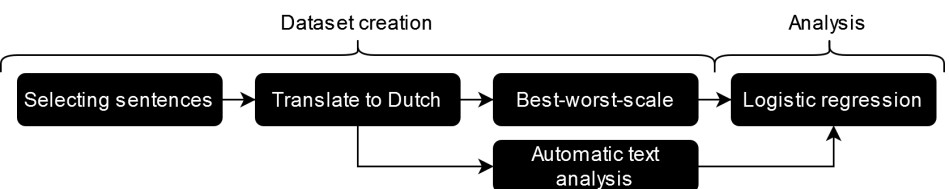

**Figure 1.** Global research design. The first part is dataset creation, and the second part is analysis.

### 3.1. Construction of the Manually Classified Dataset

Figure 2 graphically displays the dataset creation. Because this study originates from research on translating question and answering datasets, the SQUAD 2.0 dataset is chosen [38]. The SQUAD dataset contains a variety of subjects. The dataset is divided into paragraphs, questions, and answers. In May 2020, a translation request was executed at three MT cloud providers: Microsoft Azure, IBM Cloud and Google Cloud. The titles, paragraphs, questions, and answers are translated from English to Dutch.

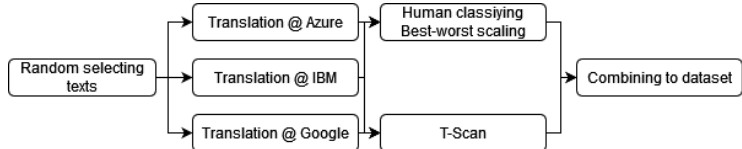

**Figure 2.** Dataset creation. First selecting the sentences, second translating the sentences from English to Dutch, third a human best-worst scale rating, fourth putting the texts into T-Scan and last combining the translations, T-Scan output and best-worst scale into one dataset.

A random selection of paragraphs from the SQUAD dataset is taken for selecting sentences. Then, the original English paragraph and the Dutch translated paragraphs are divided with a sentence splitter of the NLTK package in Python (`sent_tokenize`) [39]. The paragraphs without an equal count of sentences compared with the source text were ruled out. The reason for ruling these sentences out is that it is more difficult to select them automatically. From the rest of the sentences, 146 sentences are selected. Because the SQUAD dataset also contains questions, 67 questions were added as sentences to the dataset. These questions are randomly chosen from all the SQUAD questions.

Next, the machine-translated sentences are classified in the best-worst scaling [40]. Sentences are positioned as best translated (1) to least well translated (3). The task presented the sentences in random order, and the ids of the different sentences were hashed so that a classifying participant could never see the MT provider behind the sentence. Thereby, the annotator could give extra information about the rating process. Table 1 shows an example of the task.

As shown in Table 1, the annotators could give extra information about their best-worst scaling. In the task. 'No extra information' was default selected. Sometimes the annotators were confused or had doubts about the selection and could specify that further under the best-worst scale. They could have doubts about two or all translated sentences.

**Table 1.** Example of different machine-translated texts with the original. In this example, Google is selected as best.

| Position | Provider | Sentences | Drag & Drop |
|---|---|---|---|
| | Original | By the early 20th century balloon, or airship, guns, for land and naval use were attracting attention. | |
| 1 | Google | Tegen het begin van de 20e eeuw trokken de ballon, of het luchtschip, kanonnen voor land- en marinegebruik de aandacht. | ↕ |
| 2 | Azure | Door het begin van de 20e eeuw ballon, of luchtschip, geweren, voor land en marine gebruik waren het aantrekken van de aandacht. | ↕ |
| 3 | IBM | Door de vroege 20e-eeuwse ballon, of luchtschip, wapens, voor land en marine gebruik trok aandacht. | ↕ |

● No extra information
○ I've doubts between position 1 and 2
○ I've doubts between position 2 and 3
○ They are all equal to me

After classifying the sentences on the best-worst scale by one human rater (author), a ranking is created, visible in Table 2. Remarkably, Google translations score far better than the other providers. The same findings were also with the other annotators. Google translations are hardly scaled as worst.

**Table 2.** Number of sentences of the three MT providers selected as best (one), second, or third.

|  | Main Annotator | | | Other Annotators | | |
|---|---|---|---|---|---|---|
|  | **One** | **Two** | **Three** | **One** | **Two** | **Three** |
| Azure | 45 | 83 | 85 | 24 | 34 | 28 |
| Google | 135 | 53 | 25 | 47 | 18 | 21 |
| IBM | 33 | 77 | 103 | 15 | 34 | 37 |

To ensure the classification is correct, other annotators are added to expose a kappa score to test the correctness of the human-annotated dataset. Five different annotators checked the classifications of the first annotator. The five annotators each classified an average of seventeen translations. The translations were randomly selected for the annotators, but the random selection considered that there was not much overlap with other annotators, only with the main annotator. Again, they couldn't know which sentences were from the MT providers.

A kappa score indicates the interrater agreement between annotators and is measured by the following equation: $\kappa = (Pr(o) - Pr(e))/(1 - Pr(e))$. $Pr(a)$ is the observed agreement, the $Pr(e)$ is the expected agreement. The inter-agreement kappa score is 0.63. This score can be interpreted in different ways, namely moderate [41] or good [42]. A high agreement score in machine translation is considered to be difficult [43].

### 3.2. Analysing the Dataset

T-Scan also analyses the sentences chosen in the first part of the methods section. As mentioned, T-Scan analyses texts and outputs over 400 features in a CSV file. Not all features were filled in; many had a non-available placeholder. The outcome of the best-worst scaling task is morphed into a binary classification: when was Google best (1) and when not (0)? Google sentences are 135 times (63%) ranked as number one and 78 times (37%) ranked as second or third. The T-Scan features are balanced with the SMOTE method [44].

Figure 3 visualise the research steps for analysing the dataset globally. We choose a logistic regression because the method explains how the features operate in the model [45]. Before the logistic regression, a recursive feature elimination (RFE) method removes most of the features of the T-Scan output [46]. After RFE, some features are added back to the analysis. These were the probability features (T-Scan group nine: probability metrics), other word frequency features and word prevalence. Not all features were added because not all features had a low *p*-value ($p = 0.05$ or lower). These features are `Lem_freq_zn_log_zonder_abw`, `Hzin conj` and `Perplexiteit_bwd`. We also used an Extra Tree Classifier (ETC) to identify correlating features because, in our analysis, we noticed that the pseudo R-squared was low. Therefore, we used ETC to find more features. The fitted features were `Pv_ww1_per_zin` and `Ontk_tot_d`.

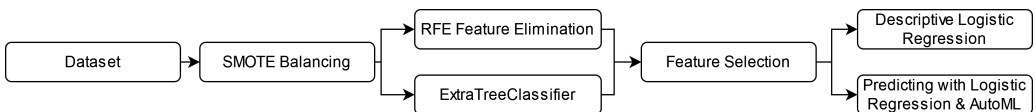

**Figure 3.** Data Analysis. First, the data are loaded; second, the dataset is SMOTE balanced. Third, in the feature selection phase with RFE and ExtraTreeClassifier, and fourth, the features are examined and tested. Fifth, a descriptive logistic regression and a prediction with logistic regression.

The software for the logistic regression comes from Statsmodels [47]. The reason for choosing a logistic regression analysis is because of the simplicity of the method. It is easily understandable which features influence the regression. Still, some features surviving the RFE had a *p*-value above 0.05 and were removed.

After the logistic regression, a prediction model is created for when Google translations are better than the other two providers. Herefore, the dataset is split into 70 per cent of the dataset for training a logistic regression. The other 30 per cent is used as a test dataset. The

training data are also balanced with SMOTE, and the features are the same at the logistic regression. The model is also logistic regression.

For further predictive examination, an AutoML test is done. For the AutoML test, TPOT is used [48]. TPOT is easy to use and works with the Scikit-Learn [49] toolbox. The library is like Scikit-Learn, a Python library. The settings in TPOT are ten generations and the whole population size (total training dataset). TPOT then searches for the optimal pipeline using genetic programming, a technique to build mathematical trees. Hence, several generations are needed to find the best pipeline.

## 4. Results

The logistic regression and prediction model will be explained in the results part.

### 4.1. Logistic Regression Analysis

Table 3 shows the logistic regression results with the balanced dataset. The table holds a pseudo $R^2$ of 0.2, and according to [50], logistic regression with a pseudo $R^2$ between 0.2 and 0.4 is well fitted. All *p*-values are lower than 0.05 and are significant features of the regression. The features are plotted individually in Figure 4. In Table 4, three example translations are given to show which characteristics will affect some logistic regression features.

**Table 3.** Logistic regression results, balanced with SMOTE.

| Dep. Variable: | y | Df Residuals: | | | | 262 |
|---|---|---|---|---|---|---|
| **Model:** | Logit | **Df Model:** | | | | 7 |
| **Method:** | MLE | **Pseudo R-squ.:** | | | | 0.2091 |
| **converged:** | True | **Log-Likelihood:** | | | | $-148.02$ |
| **Covariance Type:** | nonrobust | **LL-Null:** | | | | $-187.15$ |
| **No. Observations:** | 270 | **LLR *p*-value:** | | | | $3.124 \times 10^{-14}$ |
| | **coef** | **std err** | **z** | ***p* > |z|** | **[0.025** | **0.975]** |
| `Freq1000_inhwrd` | 2.0691 | 1.026 | 2.016 | 0.044 | 0.057 | 4.081 |
| `Lem_freq_zn_log_zonder_abw` | 0.3637 | 0.113 | 3.211 | 0.001 | 0.142 | 0.586 |
| `Hzin_conj` | $-0.6250$ | 0.258 | $-2.423$ | 0.015 | $-1.131$ | $-0.119$ |
| `Pv_ww1_per_zin` | $-0.8707$ | 0.304 | $-2.862$ | 0.004 | $-1.467$ | $-0.274$ |
| `Ontk_tot_d` | $-0.0331$ | 0.008 | $-4.281$ | 0.000 | $-0.048$ | $-0.018$ |
| `Conn_TTR` | 0.9554 | 0.336 | 2.844 | 0.004 | 0.297 | 1.614 |
| `Ww_d` | $-0.0084$ | 0.002 | $-4.456$ | 0.000 | $-0.012$ | $-0.005$ |
| `Perplexiteit_bwd` | $-0.0120$ | 0.005 | $-2.267$ | 0.023 | $-0.022$ | $-0.002$ |

Eight of the features originate from five of the feature groups of T-Scan. These groups are (1) word difficulty, with features `freq1000_inhrwd` and `Lem_freq_zn_log_zonder_abw`, (2) sentence complexity, with features `Ontk_tot_d`, `Pv_ww1_per_zin`, and `Hzin_conj`, (3) relational coherency and situation model metrics, with feature `Conn_TTR`, (4) other lexical information, with feature `Ww_d` and (5) probability metrics with feature `Perplexiteit_bwd`. Figure 4 presents the individual regression plots.

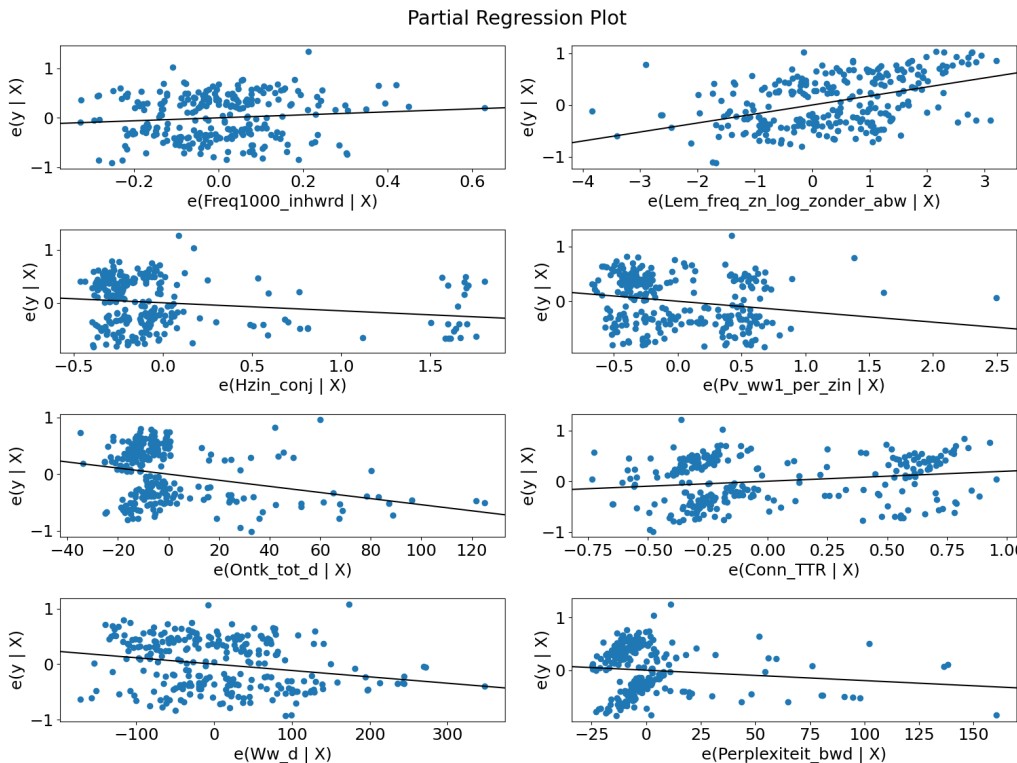

**Figure 4.** Regression plots. For each feature, it is hard to distinguish between the best scale and the other scale.

**Table 4.** Examples of when Google was classified as the worst.

| | |
|---|---|
| En: | Under the regulations of the California Constitution, no runoff election was required. |
| Nl: | Volgens de voorschriften van de grondwet van Californië waren geen verkiezingsverkiezingen vereist. |
| Exp: | In this case, the translation created a new word: verkiezingsverkiezingen. Which means election elections. It is not a real word; in this case, it negatively affected the `Lem_freq_zn_log_zonder_abw` and the `Freq1000_inhwrd` feature. |
| En: | There is also the related British Empire Medal, whose recipients are affiliated with, but not members of, the order. |
| Nl: | Er is ook de gerelateerde Medaille van het Britse Rijk, waarvan de ontvangers zijn aangesloten bij, maar niet zijn aangesloten bij, de orde. |
| Exp: | The sentence contains two times the same word: aangesloten (affiliated). First, in a positive matter, and the second time in a negative. The double word 'aangesloten' affects the `Ww_d` positive and adds another verb to the sentence. While the feature correlates negatively. |
| En: | How was the Portuguese bailout implemented? |
| NL: | Hoe is de Portugese reddingsoperatie geïmplementeerd? |
| Exp: | In this case, the preference went to the word bailout instead of reddingsoperatie (rescue operation) because the question was about some Portuguese economic event. These kinds of sentences are difficult to translate; context is essential here. In this case the feature `Lem_freq_zn_log_zonder_abw` was lower than the other two. |

### 4.1.1. Features from Word Difficulty

Feature `Freq1000_inhwrd` is the proportion of content words compared with the thousand most frequent words. The coefficient is positive. Hence, frequently used words have a positive effect on the translations.

`Lem_freq_zn_log_zonder_abw` is also a frequency-based feature. In this case, the words are lemmatised, and names and adverbs are excluded. The score is a result of the logarithm of the frequency.

### 4.1.2. Features from Sentence Complexity

Feature `Pv_ww1_per_zin` is a metric for the finite verbs at the beginning of a sentence. The coefficient correlates negatively. The reason that the feature is significant and fits in the regression is possible to the fact that 22% of the sentences were questions.

`Hzin_conjn` means the number of secondary declarative main clauses and is also negative. The negative coefficient can also be explained when a translation makes a too-long sentence of the translation. From the 25 times that Google was ranked worst, it had twelve times a higher word count than the best-scaled translation, five times there was no difference, and eight times the translation had fewer words.

Both features `Pv_ww1_per_zin` and `Hzin_conjn` are strange because most of the features are 0. However, both fitted in the logistic regression.

### 4.1.3. Features from Relational Coherency and Situation Model Metrics

`Conn_TTR` is a token type ratio for temporal, contrastive, comparative, and causal connectives and is also a positive coefficient. Words in this context are, for example, because, and, before.

### 4.1.4. Features from Other Lexical Information

Feature `Ww_d` refers to the density of verbs. It has a slightly negative coefficient. The mean of the `Ww_d` feature is at the negative group 199 and at the positive group 148. The density of verbs correlates negatively in this dataset. This is also seen in Figure 4, where most of the points in the plot are close to each other, but when the density rises, the more likely it is that it correlates negatively.

### 4.1.5. Features from Probability Metrics

`Perplexiteit_bwd` is perplexity backwards and is positively correlated to the regression. The probability coefficient only applies to content words. The feature has a *p*-value of 0.023 and is manually added to explore if it fits the model.

The mean when Google was not ranked as best is two points lower than when Google was rated as best, respectively 11.7 negatives and 13.4 when it is positive. Most of the points are close to each other, with some extreme outliers.

### *4.2. Predicting: Test Statistics*

The prediction results are divided into a logistic regression and a generated model from TPOT AutoML.

### 4.2.1. Logistic Regression

A logistic regression model predicts when the Google translation is better than the other two translations. The confusion matrix of the logistic regression is presented in the upper part of Table 5. A machine learning model is created in this part with the features shown in Table 3. A recall of 0.72 when Google was not better than other providers could be better. However, the accuracy of 0.59 leaves room for improvement. When a dataset is larger than the current one, the scores of the test statistics would probably increase. The logistic regression is insufficient to predict when Google is better than the other two translator providers.

**Table 5.** Classification report of the logistic regression model and the Gradient Boosting Classifier, generated by TPOT. The numbers in bold represent the highest score in the table.

| Classification Report: Logistic Regression | | | | |
|---|---|---|---|---|
| | **Precision** | **Recall** | **F1-Score** | **Support** |
| 0 | 0.62 | 0.45 | 0.52 | 40 |
| 1 | 0.57 | **0.72** | 0.64 | 40 |
| accuracy | | | 0.59 | 80 |
| weighted avg | 0.59 | 0.59 | 0.59 | 80 |
| Classification Report: Gradient Boosting Classifier | | | | |
| | **Precision** | **Recall** | **F1-Score** | **Support** |
| 0 | 0.68 | 0.42 | 0.52 | 40 |
| 1 | 0.58 | **0.80** | 0.67 | 40 |
| accuracy | | | 0.61 | 80 |
| weighted avg | 0.63 | 0.61 | 0.60 | 80 |

Figure 4 also gives an insight into the individual features of the model and it makes a combined model for the use case probably unsuitable.

### 4.2.2. AutoML

A pipeline was generated with the Python library TPOT, an AutoML tool. TPOT identified the Gradient Boosting Classifier as the best pipeline for predicting when Google is ranked best.

```
Best pipeline:
GradientBoostingClassifier(
    Normalizer(
        MaxAbsScaler(
            PolynomialFeatures( input_matrix , degree=2,
                include_bias=False , interaction_only=False
            )
        ),
        norm=l2
    ), learning_rate=0.1, max_depth=3,
    max_features=0.6000000000000001, min_samples_leaf=11,
    min_samples_split=14, n_estimators=100, subsample=0.55
)
```

The bottom part of Table 5 shows that the effort to optimize an ML model through AutoML does not give extraordinary results compared to the upper part. Of course, there is a recall of 0.80 for predicting when Google was better than the other providers, but the recall of when Google was worse is 0.42. Overall, the accuracy went up by 0.02 points.

## 5. Discussion and Future Work

With only eight features from T-Scan, we could relatively predict when Google was better than Azure and IBM. Of course, the accuracy stopped at 0.61, almost as high as our Kappa score. The task is difficult for a statistical model as it is for humans.

The kappa score shows that human quality estimation is not unambiguous as we hoped. With more annotators per translation, the consensus should be better determined with, for example, election [51]. In our case, we should have multiple judges to appoint the best. For the ease selecting the winning sentence, a uneven number of judges need to be applied. The downside of this is, of course, that it takes more time to rank the sentences.

The results show a simple logistic regression; most features make sense. For example, the backward metric of perplexity `Perplexiteit_bwd`, because the feature showed context between words when Google was number one compared with Azure and IBM. The test statistics show that this dataset can generate a reasonable model. However, a more significant number of data will help increase the test statistics' robustness.

Another interesting point is a different kind of data. This research only used Wikipedia data and related questions from the SQUAD 2.0 dataset. These texts are subject-oriented and written to be informative. Not all texts look like Wikipedia texts. Hence, will the same features apply to prose, poetry, news bulletins, and crowd-created messages, such as social media state utterances? Not only various kinds of texts are interesting, but also diverse kinds of source languages and target languages.

However, a multilevel logistic regression model in future work should create a better comparison between the three APIs or even with more APIs. Moreover, neural networks, support vector machines and other statistical models must be considered. Another perspective is an analysis of the text metrics between the source and target languages. What would be the critical features between the two languages to predict which API is better than others?

## 6. Conclusions

Machine translation evaluation is a challenging task. This paper gives insights into a novel and alternative approach to explain the quality of the translation without the time-consuming jobs of machine translation evaluation. Text metrics can show something about the quality and what characteristics wrong machine-translated texts have.

We expected that word probability and entropy should fit the regression, but this did not happen. The same applied to word prevalence. On the other hand, two features of word probability and one lemma frequency feature fit well into the regression.

As expected, perplexity backward correlates positively with better-written machine translations. T-Scan's readability and text metrics show insights into translation correctness. When balanced, eight features correlate with the dataset. However, the predictive model is still insufficient for an English-to-Dutch QE setting. More data are needed for model development.

**Author Contributions:** Conceptualization, C.v.T.; methodology, C.v.T.; software, C.v.T.; validation, C.v.T.; formal analysis, C.v.T.; investigation, C.v.T.; resources, C.v.T.; data curation, C.v.T.; writing—original draft preparation, C.v.T. and M.S. (Marijn Schraagen); writing—review and editing, C.v.T., F.v.D., M.S. (Marijn Schraagen), M.B. and M.S. (Marco Spruit); visualization, C.v.T.; supervision, M.S. (Marijn Schraagen), M.B. and M.S. (Marco Spruit); project administration, C.v.T.; funding acquisition, M.B. and M.S. (Marco Spruit). All authors have read and agreed to the published version of the manuscript.

**Funding:** This research was funded by P-Direkt, Ministry of the Interior and Kingdom Relations, The Netherlands.

**Institutional Review Board Statement:** Not applicable.

**Informed Consent Statement:** Not applicable.

**Data Availability Statement:** The analysis code and dataset is available on https://github.com/7083170/Readability-metrics-for-machine-translation (accessed on 26 March 2023).

**Acknowledgments:** We would like to thank the employees of P-Direkt and Utrecht University who made the interrater agreement possible for doing research.

**Conflicts of Interest:** The authors declare no conflict of interest.

## Abbreviations

The following abbreviations are used in this manuscript:

| | |
|---|---|
| MT | Machine translation |
| MTE | Machine translation evaluation |
| QE | Quality estimation |
| BLUE | Bilingual evaluation understudy |
| WMT | conference on Machine translation |
| SMT | Statistical machine translation |
| NMT | Neural machine translation |
| NER | Named entity recognition |
| RFE | Recursive feature elimination |
| ETC | Extra tree classifier |
| AutoML | Automated Machine learning |

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
