# Peer review of "Readability Metrics for Machine Translation in Dutch: Google vs. Azure & IBM"

_applsci, doi:10.3390/app13074444_

Round 1
Reviewer 1 Report
Dear Authors,
This manuscript claims to introduce a novel method to predict when a Google translation is better than other machine translations (MT) in Dutch. However, according to the evaluation metrics, the method used in this study is not an acceptable method for the purpose. In order to predict when a Google translation is better than other machine translations (MT) in Dutch, the machine learning approaches shown as Future work can be applied and a comparison study can be performed.
Author Response
Dear reviewer,
Thank you for your time and your review. I will discuss your commentary point by point:
- English language and style are fine/minor spell check required
- I checked the spelling of English in the manuscript and saw some flaws. Therefore I improved the English in the manuscript.
- Must be improved:
- Does the introduction provide sufficient background and include all relevant references?
- I improved the introduction with a clearer description that this study is an experimental setting.
- Are all the cited references relevant to the research?
- I added three new references to the research
- Are the results clearly presented?
- I discuss the results in more elaborate detail, for example, perplexity and the density of the verbs.
- Are the conclusions supported by the results?
- Indeed the conclusions needed more improvement. We added more detail about the predictive possibilities
- Does the introduction provide sufficient background and include all relevant references?
- Not applicable:
- Is the research design appropriate?
- We added more detail to the research design. Moreover, the contribution to the research is the analysis and the creation of a dataset.
- Are the methods adequately described?
- Hopefully, with more detail in the method description, the method will be more precise now.
- Is the research design appropriate?
Again, thank you for your peer review.
With kind regards,
The authors
Reviewer 2 Report
Although the authors propose an interesting research question, the experimental setting, the literature review, and the methodology are unclear in the paper.
* Concerning the T-Scan metrics, the authors generally state that the methodology provides 400 features in a CSV file in §3.2, then they describe some of those features at a very high level in §4.1, while those features are mentioned in the Related Work section. The reader expects a more cohesive presentation, where all of these considerations are stated in §2.3. Furthermore, the authors provide no evidence on how such a set of features might assess the translation's readability or quality.
* Although the authors declare that they want to assess Google, Azure (Microsoft), and IBM automatic translations, there is no explicit reference to how such technologies work. Despite those being closed under NDA, there might be some blogs or white papers/blueprints where the companies hint which are the methodologies involved in text translation. Only by doing that can the authors correlate the features associated with the black-boxed approaches and the extracted features declaring the superiority of one given approach. Given that such information is missing, no validation of the experimental result is possible.
Furthermore, the ranking methodology envisioned by the authors is flawed. as "the translations were randomly selected for the annotators, but the random selection took into account that there was not much overlap with other annotators". This seriously flaws the overall ranking scheme, as it is almost impossible to determine whether the annotators reach a consensus on the marking scheme. The authors should consider using election algorithms. Please see [1] as an example. The same problem can also be reduced into a schema alignment problem, where the alignment is between one of the three translations and the original English sentence. This approach can then be used to determine which translation was, most of the time, better than the competing ones, thus declaring the overall best translation [2,3]. Then, the authors should have applied the logistic regression (or any other kind of explainer) to determine which features predicts that one translation is good if compared to the others.
As there was little or no overlap between the translations' reviews, it is impossible to determine whether there was a consensus that Google is indeed the best competitor. As per the previous discussion, the authors should also motivate the correlation between the competitors' methodology providing correct translations with the extracted features, thus justifying the correctness of the analysis and of the proposed methodology.
Nevertheless, a more accurate assessment of the similarity between the two texts would have been done through approximate query matching, where: 1) each full sentence is represented as a dependency graph (this is done, as the translation into a graph is translingual [4]) and then the alignment between the nodes is carried out through the similarity of two terms. Tools like ConceptNet or BabelNet might be used to determine whether two tokens express the same concept in two different languages. By doing that, the authors might objectively assess the similarity between the original English text and the three translations. Then, this outcome might be calibrated upon the Mechanical Turk exploited to assess and rank the translations.
The paper might be accepted upon heavily revising the methodological and experimental setting.
[1] Areeba Umair et al. "Applications of Majority Judgement for Winner Selection in Eurovision Song Contest". https://dl.acm.org/doi/10.1145/3548785.3548791
[2] Imene Ouali et al. "Ontology Alignment using Stable Matching" https://doi.org/10.1016/j.procs.2019.09.230
[3] Tomer Sagi et al. "Non-binary Evaluation for Schema Matching" https://link.springer.com/chapter/10.1007/978-3-642-34002-4_37
[4] https://nlp.stanford.edu/software/nndep.html
Author Response
Dear reviewer,
Thank you for your time and your review. I will discuss your commentary point by point:
- English language and style are fine/minor spell check required
- I checked the spelling of English in the manuscript and saw some flaws. Therefore I improved the English in the manuscript.
- Must be improved:
- Does the introduction provide sufficient background and include all relevant references?
- I improved the introduction with a clearer description that this study is an experimental setting.
- Are all the cited references relevant to the research?
- I added three new references to the research
- Are the results clearly presented?
- I discuss the results in more elaborate detail, for example, perplexity and the density of the verbs.
- Are the conclusions supported by the results?
- Indeed the conclusions needed more improvement. We added more detail about the predictive possibilities.
- Does the introduction provide sufficient background and include all relevant references?
- Not applicable:
- Is the research design appropriate?
- We added more detail to the research design. Moreover, the contribution to the research is the analysis and the creation of a dataset.
- Are the methods adequately described?
- Hopefully, with more detail in the method description, the method will be more precise now.
- Is the research design appropriate?
- Comments and Suggestions for Authors:
- Although the authors propose an interesting research question, the experimental setting, the literature review, and the methodology are unclear in the paper.
- With the help of the new figure at the beginning of the method section and further elaboration on the methods, we hopefully addressed these described concerns.
- Concerning the T-Scan metrics, the authors generally state that the methodology provides 400 features in a CSV file in §3.2, then they describe some of those features at a very high level in §4.1, while those features are mentioned in the Related Work section. The reader expects a more cohesive presentation, where all of these considerations are stated in §2.3. Furthermore, the authors provide no evidence on how such a set of features might assess the translation's readability or quality.
- We described on purpose the features in §4.1 and not in §2.3. This is because there are more than 400 features, and we considered not elaborating on those features in §2.3 in detail. The global overview should indicate how T-Scan analyses text automatically. Readability can be displayed in different ways. In the section of Readability tooling §2.2 we described some methods of readability, such as word prevalence and word frequencies. These terms are coupled with the section of §2.2 and §2.3 with each other.
- Although the authors declare that they want to assess Google, Azure (Microsoft), and IBM automatic translations, there is no explicit reference to how such technologies work. Despite those being closed under NDA, there might be some blogs or white papers/blueprints where the companies hint which are the methodologies involved in text translation. Only by doing that can the authors correlate the features associated with the black-boxed approaches and the extracted features declaring the superiority of one given approach. Given that such information is missing, no validation of the experimental result is possible.
- With the help of the paper of Wu et al. in Google’s Neural Machine Translation System [26], we elaborate the system in more detail.
- Furthermore, the ranking methodology envisioned by the authors is flawed. as "the translations were randomly selected for the annotators, but the random selection took into account that there was not much overlap with other annotators". This seriously flaws the overall ranking scheme, as it is almost impossible to determine whether the annotators reach a consensus on the marking scheme. The authors should consider using election algorithms. Please see [1] as an example. The same problem can also be reduced into a schema alignment problem, where the alignment is between one of the three translations and the original English sentence. This approach can then be used to determine which translation was, most of the time, better than the competing ones, thus declaring the overall best translation [2,3]. Then, the authors should have applied the logistic regression (or any other kind of explainer) to determine which features predicts that one translation is good if compared to the others.
- We now take the election algorithms from the paper of Areeba Umair et al. into account in future research. I think you're right about consensus. But this is exploratory research about an alternative approach to quality estimation. More annotators are indeed necessary for the expansion of this direction.
- As there was little or no overlap between the translations' reviews, it is impossible to determine whether there was a consensus that Google is indeed the best competitor. As per the previous discussion, the authors should also motivate the correlation between the competitors' methodology providing correct translations with the extracted features, thus justifying the correctness of the analysis and of the proposed methodology.
- We expanded table 2 in the revised version. The table shows that also the other annotators chose Google as best competitor.
- Nevertheless, a more accurate assessment of the similarity between the two texts would have been done through approximate query matching, where: 1) each full sentence is represented as a dependency graph (this is done, as the translation into a graph is translingual [4]) and then the alignment between the nodes is carried out through the similarity of two terms. Tools like ConceptNet or BabelNet might be used to determine whether two tokens express the same concept in two different languages. By doing that, the authors might objectively assess the similarity between the original English text and the three translations. Then, this outcome might be calibrated upon the Mechanical Turk exploited to assess and rank the translations.
- Although the authors propose an interesting research question, the experimental setting, the literature review, and the methodology are unclear in the paper.
Again, thank you for your peer review. It was a comprehensive and good review and, in our opinion, improved the article.
With kind regards,
The authors
Reviewer 3 Report
This article provides a method for comparing machine translation amongst Google, Azure, and IBM.
The manuscript is clear and methods are described well and the problem that this article addresses is interesting.
The experimental section is very weak in this article, with very limited analysis of the datasets used. Still, I recommend publishing the manuscript.
Author Response
Dear reviewer,
Thank you for your time and your review. I will discuss your commentary point by point:
- English language and style are fine/minor spell check required
- I checked the spelling of English in the manuscript and saw some flaws. Therefore I improved the English in the manuscript.
- The experimental section is very weak in this article, with very limited analysis of the datasets used
- We elaborate the analysis in more dept. With adding more detail to the features and involve more figure 4 with the regression plots.
Again, thank you for your peer review.
With kind regards,
The authors
Reviewer 4 Report
This paper introduces a novel method to predict when a Google translation is better than other machine translations (MT) in Dutch. Instead of taking fidelity into account, this method considers fluency and readability.
The reviewer comments and concerns:
• Some sentences are too long to make readers confused, and there are also some typos and grammar errors in this paper.
• In abstract, the context of study must be added in this section and the result of this work is not clear, it must to be described briefly.
• The introduction part should include the motivation of work, it is necessary to mention the proposed contribution and defined it shortly.
• This paper lacks in-depth discussions in section 4. You must justify the effectiveness of the proposed method by giving more interpretation of the figures and tables.
• The quality of figures should be improved.
• In order to highlight the innovation of this work, it is better to cite more up-to-date references.
• Please improve the reference format and verify the number of each reference cited in the paper.
Author Response
Dear reviewer,
Thank you for your time and your review. I will discuss your commentary point by point:
- English language and style are fine/minor spell check required
- I checked the spelling of English in the manuscript and saw some flaws. Therefore I improved the English in the manuscript.
- Can be improved:
- Does the introduction provide sufficient background and include all relevant references?
- I improved the introduction with a clearer description that this study is an experimental setting.
- Are the methods adequately described?
- With more detail in the method description, the method will be more precise now.
- Are the results clearly presented?
- I discuss the results in more elaborate detail, for example, perplexity and the density of the verbs.
- Does the introduction provide sufficient background and include all relevant references?
- Comments and Suggestions for Authors:
- Some sentences are too long to make readers confused, and there are also some typos and grammar errors in this paper.
- This point is addressed at the first point in this reply.
- In abstract, the context of study must be added in this section and the result of this work is not clear, it must to be described briefly.
- Indeed, the abstract was not clear about the results and the context. The improved abstract show that our research is done in the field of quality estimation.
- The introduction part should include the motivation of work, it is necessary to mention the proposed contribution and defined it shortly.
- We added more details about our contributions to the field of quality estimation. Also we emphasize more than before about the creation of our dataset.
- This paper lacks in-depth discussions in section 4. You must justify the effectiveness of the proposed method by giving more interpretation of the figures and tables.
- We combined figure 4 with the plots more to the results and the feature descriptions.
- The quality of figures should be improved.
- We changed the plot figure with by increasing the font size
- In order to highlight the innovation of this work, it is better to cite more up-to-date references.
- Three new papers are added from 2020.
- Please improve the reference format and verify the number of each reference cited in the paper.
- We changed the papers which came from Arxiv and changed if possible to the ACL format.
- Some sentences are too long to make readers confused, and there are also some typos and grammar errors in this paper.
Again, thank you for your peer review. It was a comprehensive and good review and, in our opinion, improved the article.
With kind regards,
The authors
Round 2
Reviewer 1 Report
Dear Authors,
The reference list is not available in the revised manuscript. You must add your reference list.
You tried to revise and improve the article according to my feedback. But, as I initially argued, the logistic model is not an acceptable method for the purpose, to predict when a Google translation is better than other machine translations (MT) in Dutch. Your findings also support this situation. Therefore, I think that performing a comparison study using different ML methods will increase the quality of your study. Therefore, I recommend you perform a comparison study with the logistic model using other ML methods such as Random Forest and Support Vector Machine (not limited to these methods). In addition, the similarity rate of the abstract should be reduced.
Author Response
Dear reviewer,
Again, thank you for your time and your review. First, about the bibliographical references. I was shocked when I noticed our manuscript's references were not rendered. I worked hard to get these references good. I e-mailed the editor of this journal, and it seemed to be a fault in the systems of MDPI. In the new upload of our manuscript, I send an extra PDF to check if everything is in order.
Further, thanks to your suggestion, we added an automated machine-learning test to our improved manuscript. We used TPOT AutoML to come up with a pipeline. There are some improvements in the whole accuracy.
With your help, we reduced the similarity rate of our abstract.
Like before, I would again thank you for your time and expertise.
With kind regards,
The authors
Reviewer 2 Report
The provided version was pretty clumsy, and therefore no bibliographical references were correctly compiled. It is not clear therefore which is the methodology being used (e.g., thus including for the election mechanism, which is just stated but neither referenced or explained how this works). The reflection provided on the automated translation approaches is now provided but very marginal, as these are not linked in the discussion of the ranked results. Furthermore, the prior results table was extended with a new one, but no explanation on how the results were provided undermine the study's replicability. I appreciate the more in depth analysis of the classifier at the end of the paper, which is the only part being significantly improved. The authors should put more care in their resubmission.
Author Response
Dear reviewer,
Again, thank you for your time and your review. First, about the bibliographical references. I was shocked when I noticed our manuscript's references were not rendered. I worked hard to get these references good. I e-mailed the editor of this journal, and it seemed to be a fault in the systems of MDPI. In the new upload of our manuscript, I send an extra PDF to check if everything is in order.
We also added an automated machine-learning test to our improved manuscript. We used TPOT AutoML to come up with a pipeline. There are some improvements in the whole accuracy.
With kind regards,
The authors
Reviewer 4 Report
The authors have responded satisfactorily to all the reviewer’s concerns. They have made a large number of significant modifications to their paper to improve its quality: adding the suggested modifications in abstract and introduction, improving the paper structure to make it more understandable, improving the results. Therefore, I recommend the acceptance of the paper.
Author Response
Dear reviewer,
Thank you for your time and your review.
With kind regards,
The authors
Round 3
Reviewer 1 Report
The paper has a few important drawbacks that prevent me from recommending its acceptance:
1- You should provide more detailed information about the process of the TPOT AutoML model applied in your article under subheading 4.2.2. (Tree-based Pipeline Optimization Tool, or TPOT for short, is a Python library for automated machine learning, etc.)
2- The subheading title in 4.1 should be renamed as “Logistic regression analysis results”
3- The abstract should more clearly include the comparison results of the TPOT AutoML and the logistic model.
4- The subheading title in 4.2.1 should be renamed as “Logistic regression”
5- The sentence in Lines 49 and 50 which “Thereby the prediction possibilities of such a model are elaborated with a logisci regression model and an automated machine learning model (AutoML).” should be rechecked and typos should be corrected.
6- In the introduction, the purpose should be explained more clearly.
7- The manuscript should be proofread again by an expert whose native language is English professionally.
Author Response
Dear reviewer,
Thank you again for this extensive feedback, and again, we think these points helped us to improve our paper. Point by point, we elaborate on your feedback:
- You should provide more detailed information about the process of the TPOT AutoML model applied in your article under subheading 4.2.2. (Tree-based Pipeline Optimization Tool, or TPOT for short, is a Python library for automated machine learning, etc.)
- We added detailed information about the TPOT process in the end of the Method section (lines 246, pages 6-7). We added also a brief but more detailed introduction to the Results section in subsection 4.2.2 (lines 313-315, page 10).
- The subheading title in 4.1 should be renamed as “Logistic regression analysis results”
- The subheading is renamed "Logistic Regression Analysis" (line 250, page 7).
- The abstract should more clearly include the comparison results of the TPOT AutoML and the logistic model.
- We indeed changed the results and wrote the results of the AutoML and logistic model down (lines 7-9, page 1).
- The subheading title in 4.2.1 should be renamed as “Logistic regression”
- The subheading is renamed "Logistic Regression" (line 250, page 7).
- The sentence in Lines 49 and 50 which “Thereby the prediction possibilities of such a model are elaborated with a logisci regression model and an automated machine learning model (AutoML).” should be rechecked and typos should be corrected.
- We changed the old sentences to a better version (lines 50-53, page 2)
- In the introduction, the purpose should be explained more clearly.
- We added the purpose explicitly now between rule 37 and rule 40 (page 1).
- The manuscript should be proofread again by an expert whose native language is English professionally.
- Due to time, we skipped this, but we will do this in the camera version. We used Grammarly Premium to find the last spelling mistakes.
Once more, thank you for your expertise, pointwise feedback and time.
Sincerely,
The authors
Reviewer 2 Report
Now it is clearer that the election mechanism is put as a future work rather that being implemented in the current system. There are still a few typos (logistic regression in a subsection where the first letter is not capitalised). Methodology has to be improved in their future work.
Author Response
Dear reviewer,
Thank you again for this extensive feedback, and again, we think these points helped us to improve our paper. Point by point, we elaborate on your feedback:
- Now it is clearer that the election mechanism is put as a future work rather that being implemented in the current system. There are still a few typos (logistic regression in a subsection where the first letter is not capitalised).
- Thanks to your feedback about capitalised headers, we uniformed the headers in our paper with title case.
- Methodology has to be improved in their future work.
- For the election part in our future work we added how should use majority elections for selecting the best sentences (lines 339-341, page 10).
Once more, thank you for your expertise, pointwise feedback and time.
Sincerely,
The authors